# Scaling Laws Revisited: Modeling the Role of Data Quality in Language Model Pretraining

**Anirudh Subramanyam**[1], **Yuxin Chen**[2], **Robert L. Grossman**[1,2,3]

[1]Center for Translational Data Science
[2]Department of Computer Science
[3]Department of Medicine
University of Chicago, Chicago, IL 60637, USA
`{anisub,chenyuxin,rgrossman1}@uchicago.edu`

## Abstract

Scaling laws for language model training traditionally characterize how performance scales with model size and dataset volume. Prior work has explored architecture variants and data treatments such as dataset filtering and noise injection in language model pretraining; however, these studies have not formalized data quality within a principled scaling law. We introduce a dimensionless data-quality parameter $Q$, and propose a quality-aware scaling law extending the Chinchilla framework to predict loss as a joint function of model size, data volume, and data quality. The law is motivated by an effective-sample-size and information-theoretic view of noisy or redundant corpora, and it admits two practical estimators for $Q$: (i) a *corruption rate* proxy and (ii) a *deficiency* measure. Through synthetic experiments in neural machine translation and autoregressive modeling—where we systematically control data quality via multiple levels of noise injection variation—we show that loss scales predictably with data quality and that higher-quality data can substantially reduce model size and hence compute requirements. Our results demonstrate a sublinear decay of effective data with quality and robustness to moderate data corruption; out-of-sample evaluations further validate the predictive form of the law. Unlike prior empirical analyses, our work establishes an explicit, generalizable law for data quality, offering concrete guidance for balancing data curation effort and model scale in large-scale pretraining.

## 1 Introduction

It is well understood in large-language model (LLM) training that both the amount of training data and its quality influence pretraining loss. Yet most existing scaling laws are formulated solely in terms of dataset size, remaining agnostic to quality (Henighan et al., 2020; Ghorbani et al., 2021; Ivgi et al., 2022; Sorscher et al., 2022; Alabdulmohsin et al., 2022; Caballero et al., 2023; Tao et al., 2024; Wu et al., 2025). This leaves a gap between common intuition–"cleaner data trains better models"–and the quantitative laws that guide large-scale training.

The importance of such a quantitative framework is especially clear in specialized domains–business, scientific, or medical applications–where training corpora are limited in quantity and heterogeneous in quality. With the same compute budget, outcomes may differ drastically depending on how clean and representative the data are. There have been quite a number of studies to classify data quality among multiple "dimensions" (Wang and Strong, 1996; Fox et al., 1994; Sidi et al., 2012; ISO/IEC, 2008). While the exact taxonomy varies, the core dimensions typically include: accuracy, completeness, consistency, timeliness, uniqueness, validity. The focus of these studies is usually datasets containing structured data that is clean and curated for a certain task or purpose. In contrast, when collecting data to pretrain large language models, large amounts of "found" data are collected and cleaned, usually by removing duplicate data and obviously "poor" quality data.

To reason about how such imperfect corpora affect performance, we need a simpler abstraction of quality. In this work, we introduce a single dimensionless parameter $Q \in (0, 1]$ characterizing the usable information in a corpus. A value of $Q = 1$ represents fully clean and representative data,

while smaller $Q$ values reflect increasing corruption or redundancy. As discussed below (Section 3), $Q$ can be estimated via *corruption rates* or a more general *deficiency* measure, and serves as a proxy for the fraction of effective samples. We incorporate this abstraction into scaling laws through a term of the form $B/(D^\beta Q^\gamma)$ (Equation 2), linking model performance directly to both the quantity and the quality of data.

Our main contributions are:

- We propose a *theoretically grounded* quality-aware scaling law that augments the classical Chinchilla form by introducing a dimensionless quality parameter $Q$. The law predicts loss as $L(N, D, Q) = A/N^\alpha + B/(D^\beta Q^\gamma) + E$, capturing the interplay between model size, data volume, and data quality.

- We derive the scaling law from an *effective sample size* perspective and information-theoretic arguments, and we provide two simple estimators for $Q$ based on corruption rate and data deficiency. We show that under natural assumptions these estimators lead to the form $D_{\text{eff}} = D\,g(Q)$ with $g(Q) \approx Q^\gamma$, recovering the proposed law.

- We conduct controlled experiments on neural machine translation and causal language modeling to *illustrate* the law's utility. These results demonstrate that loss scales predictably with our quality parameter and that higher-quality data can compensate for smaller models. This is particularly relevant for specialized models in many real-world domains, such as business, scientific or medical applications.

## 2 RELATED WORKS

**Data quality in information systems.** Data quality has long been studied in information systems, where datasets are evaluated along multiple dimensions—accuracy, completeness, consistency, timeliness, uniqueness and validity—to ensure that structured records support downstream tasks. Note that what call completeness is also called coverage and closely to what is usually called diversity. A consistent finding from this literature is that diverse imperfections degrade the utility of data and therefore motivate extensive curation practices (Batini et al., 2009). Large-scale pretraining, however, relies on corpora assembled from the web. Here, quality varies widely across sources, and subtle forms of corruption or deficiency persist despite deduplication or heuristic filtering. Nonetheless, the key insight from information systems continues to hold: the quality of training data directly influences the performance of learned models.

**Data quality in large-scale pretraining.** In the era of LLMs, quality has become a first-class design concern. Major corpora such as The Pile (Gao et al., 2020), RefinedWeb (Penedo et al., 2023), and Dolma (Soldaini et al., 2024) highlight different philosophies of data construction, from assembling diverse sources to aggressively filtering and refining web text. Deduplication strategies (Lee et al., 2022) and contamination checks (Magar and Schwartz, 2022) have been shown to substantially affect evaluation reliability. Empirical analyses further suggest that careful filtering yields performance gains comparable to scaling compute, therefore implying the nontrivial role of quality in driving LLM performance.

**Classical scaling laws and their limitations.** Alongside these developments, foundational studies on scaling laws formalized how performance scales with model size, dataset size, and compute. Kaplan et al. (2020) showed that loss decreases predictably as models and datasets grow, while Hoffmann et al. (2022) refined these results by introducing compute-optimal training. Despite their influence, these existing scaling laws generally assume that the underlying data is of fixed quality. In real-world corpora, noise, redundancy, or domain imbalance systematically distort scaling behavior, highlighting the need to explicitly account for quality.

**Data quality and scaling laws.** Several recent works have begun to address this gap. Bansal et al. (2022) study scaling laws for neural machine translation, and show that while test loss scales predictably with dataset size, architectural choices and moderate noise primarily shift the scaling curves without changing the exponent. Goyal et al. (2024) argue that data filtering is not compute-agnostic: the value of higher-quality data depends on the available compute budget, since reusing small clean datasets leads to diminishing returns. Li et al. (2024) assess data quality by comparing

perplexity gaps between models of different sizes, theoretically linking their metric to inverse scaling laws. These studies highlight the importance of data quality but stop short of providing a unified predictive law that couples quality with model size and dataset size. By introducing a dimensionless quality parameter $Q$ into scaling laws, our work formalizes the influence of data quality and offers predictive guidance for building smaller, high accuracy models in specialized domains.

Beyond these empirical analyses, several recent studies refine scaling laws from complementary perspectives. Chang et al. (2024) propose *parameter-constrained* scaling laws that combine text diversity and syntheticity, revealing that effective training tokens depend on both data quantity and diversity. On the theoretical front, Bahri et al. (2024) present a unifying theory of neural scaling laws, distinguishing variance-limited and resolution-limited regimes and connecting scaling exponents to generalization error. Although these contributions deepen our understanding of scaling, they do not incorporate an explicit scalar quality parameter. Our work complements this literature by modeling data quality via $Q$, linking it to effective sample size and generalization theory, and providing a unified scaling law that accounts for quality.

## 3 DATA QUALITY MEASURES

To incorporate data quality into scaling laws, we first need a formal definition of $Q$. In this section we introduce two approaches for mapping a dataset $\omega$ to a scalar $Q(\omega) \in (0, 1]$, with larger $Q$ indicating higher quality. These definitions capture different sources of imperfection—token corruption versus more general deficiency—and will later allow us to connect $Q$ to effective sample size and generalization theory. The precise choice of $Q$ is application-dependent; the key requirement is that $Q$ degrades smoothly under corruption, providing a proxy for the usable information in a dataset.

### 3.1 DATA CORRUPTION RATE

One of the simplest measures of data quality is what is sometimes called the *data corruption rate*, which we will denote CR. We assume $0 \leq \text{CR} < 1$.

**Definition 1.** *Let $\omega$ be a dataset with data corruption rate CR. We define the data corruption rate data quality by*

$$Q(\omega) = 1 - \text{CR}.$$

As an example, if we have a dataset consisting of $D$ tokens and $10\%$ are corrupted, then we would say that the $\text{CR} = 10\%$ and the data quality $Q$ is $90\%$. To simplify our formulas, we assume that the data corruption rate is strictly less than $1$, so that $Q$ is strictly greater than $0$. Note that standard sampling methods (e.g. Deming (1966)) can be used to estimate data corruption.

### 3.2 DATA DEFICIENCY MEASURES

We introduce a dimensionless quantity $\Delta$ call the *data deficiency*, which quantifies the lack of quality in a dataset $\omega$. We assume that the data deficiency has the following properties:

1. **Positivity.** $\Delta(\omega) \geq 0$, for all datasets $\omega$

2. **Continuity.** $\Delta(\omega)$ is a continuous function of the dataset $\omega$

3. **Additivity.** If $\omega_1$ and $\omega_2$ are independent datasets, and $\omega$ is the union of $\omega_1$ and $\omega_2$ than $\Delta(\omega) = \Delta(\omega_1) + \Delta(\omega_2)$.

**Additive noise model.** Data deficiency can be measured and quantified in many ways. One of the simplest models is to model data deficiency using an *additive noise* model, where each data measurement $\xi_i$, for $i = 1, \ldots$, has the form

$$\xi_i = x_i + \epsilon_i,$$

where $\epsilon_i$ is a noise term. We assume that $\epsilon_i \geq 0$, and that the data quality as measured by the data deficiency worsens as $\epsilon_i$ increases. Note that properties (1)—(3) above hold when we model data quality using the additive noise model. We now add a fourth assumption.

4. **Maximum quality.** In the additive noise model, we assume that the noise term $\epsilon_i$ captures all the noise, so that data quality is maximized when $\epsilon_i = 0$, for all $i$.

Note that the data deficiency $\Delta$ can grow arbitrarily large with this model, so that $0 \leq \Delta(\omega) < \infty$, for a dataset $\omega$.

**Definition 2.** *We define the data quality induced by data deficiency $Q(\omega)$ of a dataset $\omega$ by*

$$Q(\omega) = \exp\left(-\Delta(\omega)\right).$$

# 4 QUALITY-AWARE SCALING LAWS

A widely used empirical scaling law for LLMs (also known as the Chinchilla Scaling Law by Hoffmann et al. (2022)) is

$$L(N, D) = \frac{A}{N^\alpha} + \frac{B}{D^\beta} + E,$$

where $N$ is the number of training parameters, $D$ is the number of training tokens, and $L(N, D)$ is the pretraining loss. This equation was estimated empirically from experimental studies involving over 400 LLM, with the number of parameters ranging from 70M to 16B and the number of training tokens ranging from 5B to 400B. Here $A$, $B$, $E$, $\alpha$, and $\beta$ are empirically estimated constants, with $E$ the minimal loss.

For example, one formulation studied in Hoffmann et al. (2022) is:

$$L(N, D) = \frac{406.4}{N^{0.34}} + \frac{410.7}{D^{0.28}} + 1.69. \tag{1}$$

In large-scale training, improvements such as filtering out low-quality data or introducing better training objectives have yielded improved performance beyond what naive scaling might predict. For instance, filtering training data to remove noise has been observed to effectively improve the scaling behavior, enabling models to achieve lower loss with the same compute. This indicates that not all tokens are equal: a billion high-quality tokens may be far more valuable than a billion noisy or redundant ones. The Chinchilla analysis already hinted that many models are *undertrained* given their size, pointing toward the need for either more data or better data. This motivates us to extend scaling laws with explicit parameters for data quality.

## 4.1 EFFECTIVE SAMPLE SIZE

We now formalize when data quality $Q$ leads to a scaling law of the form $D^{-\beta}Q^{-\gamma}$. The key idea is that poor data quality (e.g. data corruption or large data deficiency) reduces the *effective sample size* of the dataset. We formalize this with:

**Definition 3.** *Let $g$ be a monotone function $g : [0, 1] \rightarrow \mathbb{R}_+$ with $g(1) = 1$. Then the effective sample size of $D$ associated with the link function $g$ is*

$$D_{\text{eff}} := D\,g(Q).$$

**Assumption 1** (Effective sample size factorization). *There exists a monotone function $g : [0, 1] \rightarrow \mathbb{R}_+$ with $g(1) = 1$ such that the expected excess loss satisfies*

$$L_N(D, Q) \approx \frac{B}{D_{\text{eff}}^\beta} = \frac{B}{(D \cdot g(Q))^\beta}.$$

*where $L_N(D, Q)$ denotes the loss for a given parameter size $N$.*

This assumption is consistent with classical results: in PAC learning with random classification noise of rate $\eta < 1/2$, the sample complexity inflates by a factor $(1 - 2\eta)^{-2}$ (Angluin and Laird, 1988; Kearns, 1998); in regression with additive Gaussian noise of variance $\sigma^2$, effective samples scale with the signal-to-noise ratio $\text{SNR} = 1/\sigma^2$ (Tsybakov, 2009); and in information theory, channel capacity arguments show that mutual information is reduced by a multiplicative factor depending on the corruption level (Cover and Thomas, 2006). In the following two lemmas (proof deferred to the appendix), we show that the usable sample size can be written as $D_{\text{eff}} = D \cdot g(Q)$ with $g(Q)$ well-approximated by a power law $Q^\gamma$ over practical ranges of $Q$:

- $\gamma \approx 1$ for signal-to-noise ratio like token noise (Lemma 1)
- $\gamma \approx 2$ for symmetric label noise (Lemma 2)

**Lemma 1** (Effective sample size under additive Gaussian noise). *Consider i.i.d. observations of the form $y_i = f(x_i) + \epsilon_i$, $\epsilon_i \sim \mathcal{N}(0, \sigma^2)$, for $i = 1, \ldots, D$. Then the Fisher information contributed by each observation is proportional to $1/\sigma^2$, and the total Fisher information satisfies*

$$\mathcal{I}_D \ \propto \ \frac{D}{\sigma^2}.$$

*Equivalently, the $D$ noisy samples carry the same information as $D_{\text{eff}} = D \cdot (1/\sigma^2)$ noise-free samples.*

We define the deficiency as

$$\Delta = \ln \frac{\sigma^2}{\sigma_0^2}, \qquad Q = e^{-\Delta} = \frac{\sigma_0^2}{\sigma^2}.$$

where $\sigma_0^2$ is baseline noise (i.e. e.g. noise not modeled by $\Delta$). Without loss of generality, we may assume $\sigma_0 := 1$. In this formulation, datasets noisier than the reference ($\sigma^2 > \sigma_0^2$) have $Q < 1$, while "clean" datasets ($\sigma^2 = \sigma_0^2$) have $Q = 1$. The effective sample size then scales as

$$D_{\text{eff}} = D \cdot Q^\gamma$$

and taking $\gamma = 1$ recovers the classical $D \cdot \text{SNR}$ scaling.

**Lemma 2** (Effective sample size under random classification noise). *Consider binary classification with labels $Y \in \{0, 1\}$ and i.i.d. examples $(X, Y)$. Suppose labels are corrupted by* random classification noise *(RCN) with flip rate $\eta \in [0, 1/2)$, i.e., we observe $\tilde{Y} = Y$ w.p. $1 - \eta$ and $\tilde{Y} = 1 - Y$ w.p. $\eta$, independently of $X$. For any classification-calibrated surrogate loss admitting an unbiased noise-corrected estimator (e.g., logistic/hinge via the standard correction), the variance of the per-example corrected loss inflates by a factor proportional to $(1 - 2\eta)^{-2}$. Consequently, generalization bounds and excess-risk rates scale as if the number of clean samples were*

$$D_{\text{eff}} = D \cdot (1 - 2\eta)^2,$$

*i.e., the effective sample size is reduced by $(1 - 2\eta)^2$ under RCN.*

Lemma 2 formalizes the intuition that symmetric label noise reduces usable information multiplicatively, yielding $D_{\text{eff}} = D \cdot g(Q)$ with $g(Q) = (2Q - 1)^2$. In other words, if we define data quality via the corruption rate as $Q = 1 - \eta$ (cf. Section 3), then $(1 - 2\eta)^2 = (2Q - 1)^2$. Over practical high-quality regimes $Q \in (1/2, 1]$, the factor $(2Q - 1)^2$ is well-approximated by a local power law in $Q$ with exponent $\gamma \approx 2$, so that an effective-sample model $D_{\text{eff}} = D \cdot Q^\gamma$ with $\gamma \approx 2$ is justified for RCN.

## 4.2 Quality-Aware Scaling Law

We incorporate the data quality $Q$ and introduce the following quality-aware scaling law:

**Definition 4** (Quality-Aware Scaling Law).

$$L(N, D, Q) = \frac{A}{N^\alpha} + \frac{B}{D^\beta Q^\gamma} + E, \tag{2}$$

*where $Q$ denotes data quality and $\gamma$ is a empirically measured parameter.*

Note that for the highest quality data with $Q = 1$, this reduces to the standard Chinchilla scaling law, and as the quality of the data decreases, the loss increases, as is expected. On the other hand, as the quality of the data goes up, the amount of the data required to achieve the same loss goes down.

The following corollary formalizes how the proposed form arises under mild regularity conditions:

**Corollary 1.** *Suppose that $g(Q)$ is regularly varying at $Q = 1$, i.e. $g(Q) = c\,Q^\gamma(1 + o(1))$ as $Q \to 1$. Then the quality-aware scaling law (Definition 4) holds under Assumption 1.*

We next give an information-theoretic perspective that leads to the same law:

**Proposition 1** (Information-theoretic justification). *Let $X$ denote clean tokens, $\tilde{X}$ their corrupted versions produced by a memoryless channel $\mathcal{C}_Q$ parameterized by quality $Q \in (0, 1]$, and let $Z$ be the learned representation used for prediction. Suppose corruption reduces usable information multiplicatively,*

$$I(\tilde{X}; Z) = \rho(Q)\, I(X; Z),$$

*with $\rho(1) = 1$, $\rho(Q)$ monotone, and locally $\rho(Q) \approx c\, Q^\gamma$ as $Q \to 1$. Here $I(X; Z)$ denotes mutual information as usual. If the data-dependent loss scales as*[1]

$$L_D \propto \frac{1}{\big(D \cdot I(\tilde{X}; Z)\big)^\beta},$$

*then the quality-aware scaling law takes the form*

$$L(N, D, Q) \approx \frac{A}{N^\alpha} + \frac{B}{D^\beta Q^\gamma} + E,$$

*after reparameterization of constants.*

The proof of Proposition 1 is deferred to the appendix. For a binary symmetric channel with flip rate $\eta$, $\rho(Q)$ is proportional to the capacity $1 - H_2(\eta)$ with $Q = 1 - \eta$; over practical ranges this is well-approximated by $Q^\gamma$ with $\gamma \approx 2$. In regression with additive Gaussian noise, $\rho(Q)$ is proportional to the signal-to-noise ratio, yielding $\gamma \approx 1$. Both cases recover the effective sample size view $D_{\text{eff}} = D \cdot \rho(Q)$.

## 4.3 COMPARISON WITH OTHER QUALITY-AWARE SCALING LAWS

Table 1: Relating different realizations of the data deficiency term $\Delta(\omega)$ to the literature

| Definition of $\Delta(\omega)$ | Interpretation of Data Deficiency | Paper |
|---|---|---|
| $\mu_1 E + \mu_2 \frac{1}{F} + \mu_3 G + \mu_4 H$ (equation 3) | Decomposition into noise ($E$), inverse coverage ($1/F$), and repeated-data measure ($G$), and syntheticity ($S$). | **Ours** |
| $\mu_2 \operatorname{Dis}(\rho)$ | Cluster-based density $\rho$, with $\operatorname{Dis}(\rho) = -\ln(1 - \rho)$. Small $\rho \to$ high-quality; large $\rho \to$ low-quality. | Chen et al. (2025) |
| $\mu_2 \operatorname{DR}(\omega) + \mu_4 S(\omega)$ | Compression-based density (compressibility as proxy for diversity) $\operatorname{DR}(\omega) = \operatorname{CR}^{-1}(\omega)$ and $\operatorname{CR} = \frac{\text{size}}{\text{compressed size}}$, and data syntheticity $S(\omega)$. Higher DR, higher $S \to$ lower quality. | Chang et al. (2024) |
| $\mu_3 \frac{k}{\tau}$ | Declining marginal value of data with repeated exposure over $k$ epochs; linear–exponential saturation depending on $\tau$. Models over-training on repeated data. | Goyal et al. (2024) |

To relate our data dispersion to the literature, assume that the data deficiency is of the form

$$\Delta(\omega) = \mu_1 E + \mu_2 \frac{1}{F} + \mu_3 G + \mu_4 H, \tag{3}$$

where $E$ quantifies the noise in the data, $F$ quantifies the coverage/diversity of the data, $G$ quantifies the amount of repeated data, and $H$ quantifies the loss in data quality due to synthetic data. As shown in Table 1, by assuming particular forms for each of these terms, we recover the data quality scaling laws in Chen et al. (2025), Goyal et al. (2024) and Chang et al. (2024).

---

[1]This assumption is motivated by information-theoretic generalization bounds (Xu and Raginsky, 2017; Bu et al., 2020), which relate generalization error to the mutual information between training data and learned representations or weights, as well as by the Information Bottleneck principle (Tishby et al., 1999).

## 5 EXPERIMENTAL STUDIES

To establish the quality aware scaling laws, we train decoder only language models across two tasks, neural machine translation (NMT) and causal language modeling (CLM). To capture the effects of quality and dataset size we train on 3 different dataset volumes and 7 degrees of quality. We use subsets of Paracrawl v8 (Bañón et al. (2020)) and C4 (Dodge et al. (2021)) as our training data for the translation and language tasks respectively. See Table 2. We optimize cross-entropy loss averaged over the relevant context and report the same on held out test datasets to measure the effectiveness of our quality aware scaling law for predicting the loss on out of sample data (Table 3. We use the parametric loss fitting approach described in Hoffmann et al. (2022) to find our parameters using two methods, Least Squares and Huber (Huber, 1992). The full details of the experiments, and model training details can be found in the appendix.

### 5.1 TASK DESCRIPTION

#### NEURAL MACHINE TRANSLATION

Our first task is English to German translation using the Paracrawl v8 English to German dataset (Bañón et al., 2020). We filter the raw data with min hash deduplication, language ID filtering with a 0.8 threshold and length filtering ($\leq 258$) leaving $\sim 101M$ sentence pairs. We use a 8L GPT Neo model with a hidden size of 1024 with approximately $\sim 133M$ params. We also train our own BPE tokenizer with a vocabulary size of 32000. We train 3 replicates for each combination of the 3 datasets of size 500K, 1M and 2M sentence pairs and 7 quality levels for a total of 63 experiment runs. We train using AdamW with a maximum learning rate of 5e-4 and cosine decay with a 20% warmup ratio. The full model configuration and training hyper-parameters are available in the appendix.

#### CAUSAL LANGUAGE MODELING

For the CLM objective we train our models on a subset of C4 (en) (Dodge et al., 2021). We run our pretraining experiments on a 8L Llama 3 (Grattafiori et al., 2024) model with a hidden size of 512 and a context length of 2048. We do not use ROPE scaling. We employ random data truncation on the base dataset to sample within the context length. We also employ the pre-trained Llama-3.2-1B tokenizer. We sample datasets of three sizes; 100M , 1B and 10B tokens and train for a single epoch using fused AdamW (Loshchilov and Hutter, 2019) with a maximum learning rate of 1e-3 with cosine decay and a 10% warm-up ratio. We duplicate our experiment setting to match a total of 63 runs. The full model configuration and training hyperparameters are available in the appendix.

### 5.2 SYNTHETIC NOISE AND DATA SAMPLING STRATEGY

To simulate the different levels of data quality we add synthetic iid noise to the base dataset. The base dataset after pre-processing is assumed to have quality, $Q = 1.0$. We vary the quality of a dataset by increasing/decreasing the fraction of samples that are perturbed with synthetic noise. For example, if 25% of all samples are perturbed, then the quality of that dataset is considered to be 0.75. We use two noise models, one for each task. For NMT, we randomly set 50% of all non-special tokens in a selected sample to pad tokens, we do not discriminate between source and target here. For CLM, we randomly swap 50% of all non-special tokens with valid non-special tokens from the tokenizer vocabulary.

### 5.3 DATA SAMPLING STRATEGY

We employ a nested subset sampling strategy that guarantees noise and sample monotonicity. $S = \{s_1, s_2, s_3, \ldots, s_n\}$ is the base experiment dataset with $n$ unique samples. We draw from $S$ thrice using different seeds to create working datasets $S_{w1}$, $S_{w2}$ and $S_{w3}$. Each working dataset $S_{wi}$ is then sampled to produce datasets of different sizes. Each of these different sized datasets are then perturbed to various degrees to create the noised variants at that dataset size.

For example, in our CLM experiments the dataset volumes are $T=\{0.1, 1, 10\}$ billion (B) tokens. We first draw thrice from C4 to get our 3 working datasets $S_{w1}$, $S_{w2}$ and $S_{w3}$ of size 40M samples or

Table 2: Estimated Parameters in Quality-Aware Scaling Law

| TASK | METHOD | B | $\beta$ | $\gamma$ | E |
|------|--------|---|---------|----------|---|
| NMT | Least Squares | 166.568727 | 0.262933 | 0.185135 | 0.146998 |
| NMT | Huber | 139.602744 | 0.250067 | 0.173161 | 0.066539 |
| CLM | Least Squares | 1428.225931 | 0.395142 | 0.388678 | 3.439888 |
| CLM | Huber | 1441.505289 | 0.395859 | 0.400657 | 3.439047 |

Table 3: Estimated Parameters on Unseen Data - CLM

| TASK | METHOD | B | $\beta$ | $\gamma$ | E |
|------|--------|---|---------|----------|---|
| CLM (ours) | Least Squares | 1428.225931 | 0.395142 | 0.388678 | 3.439888 |
| CLM (unseen) | Least Squares | 1589.071797 | 0.396787 | 0.332273 | 4.551611 |
| CLM (ours) | Huber | 1441.505289 | 0.395859 | 0.400657 | 3.439047 |
| CLM (unseen) | Huber | 1427.299279 | 0.390546 | 0.336753 | 4.540009 |

$\simeq$16B tokens each (Using the Meta Llama-3.2-1B tokenizer, we find that on average the "en" subset of C4 has approximately 400 tokens per sample). We ensure that for each working dataset $S_{wi}$, we sample subsets of the form, i.e. $S_{0.1B} \subseteq S_{1.0B} \subseteq S_{10.0B}$. At each subset level, we then compose 7 different dataset variants with $\eta = \{0, 10, 20, 25, 30, 40, 50\}$ percent noised samples (where $\eta$ denotes the noise percentage), meaning that quality $Q = \{1.0, 0.9, 0.8, 0.75, 0.7, 0.6, 0.5\}$.

## 5.4 EXPERIMENTAL SETUP

We make use of multiple compute sources in this work. NMT experiments upto a dataset volume of 1M sentence pairs were conducted on an A100 node with 4, 80GB GPUs. All other experiments are run on 2 Hopper nodes with 4, 141GB GPUs each. We do not use multi-node or multi-GPU training. We utilise the GPU counts to parallelize experiment runs.

## 5.5 MAIN RESULTS

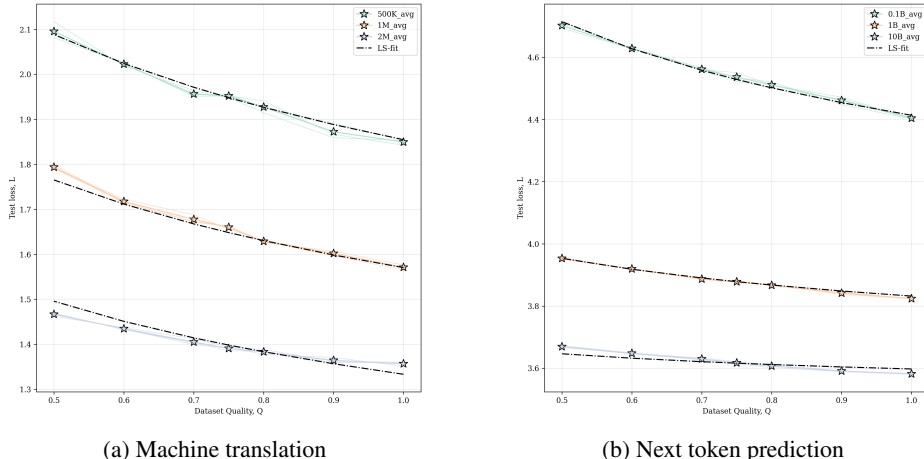

(a) Machine translation

(b) Next token prediction

Figure 1: Test loss as data quality varies from 0.5 to 1 for a) machine translation and b) next token prediction.

**Fitting the quality-aware scaling law.** Our main results are summarized in Table 2, which estimates $B$, $\beta$, $\gamma$, and $E$, and Figure 1, which shows the test loss as $Q$ varies from 0.5 to 1.0. Pre-

dictably, we find that test loss $L$ decreases as we increase the volume of data $D$ and increase the fraction of high quality samples $Q$. Table 3 shows us that using our pre-trained models to evaluate loss on unseen data also follows a scaling law similar to our fit on in distribution data, thus showing evidence for scaling generalization. To further illustrate trade-offs between $D$ and $Q$, Figure 2 presents iso-loss contours, showing how data quality improvements can substitute for increased dataset size at fixed model capacity.

Interestingly, the estimated exponents for data quality, $\hat{\gamma}$, are significantly less than one: $\hat{\gamma} \approx 0.173$ for NMT and $\hat{\gamma} \approx 0.401$ for CLM (with Huber estimation). This indicates that the effective dataset size decays sublinearly with quality, i.e., models are more robust to moderate corruption than predicted by simple effective sample-size theories from PAC learning or channel-capacity analysis, which typically suggest $\gamma \geq 1$. We hypothesize that this robustness arises from redundancy in natural language data, where even partially corrupted samples carry useful contextual information (e.g., syntax, alignment, or co-occurring clean tokens). The higher $\gamma$ in CLM compared to NMT suggests that autoregressive language modeling is more sensitive to token corruption, whereas NMT can leverage cross-sequence redundancy to mitigate the impact of noise. Another plausible hypothesis is that in the NMT noise model (padding half the tokens in a noised sample), the corrupted samples are not completely useless: alignment and context still leak enough information. In CLM (token swaps), the corruption is harsher for distributional modeling, since the noised tokens add entropy and can mislead local dependencies. Hence a larger $\gamma$.

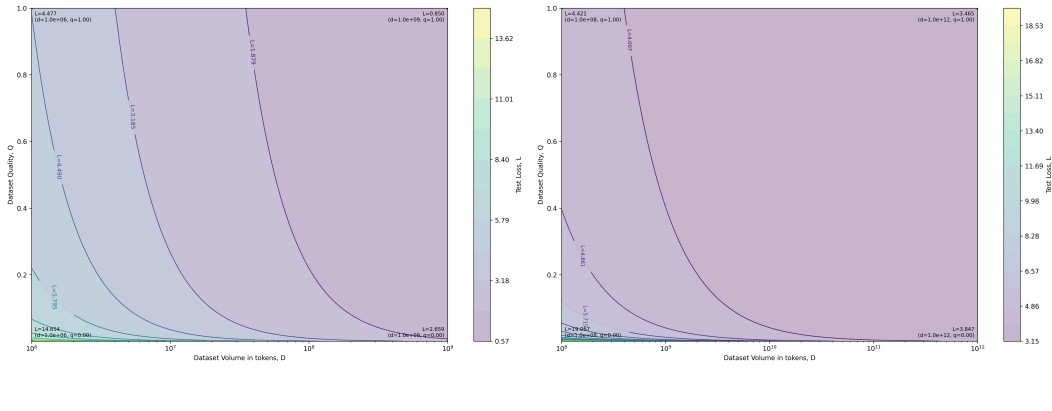

(a) Machine translation contours  (b) Next token prediction contours

Figure 2: Iso loss contours for machine translation (a) and next token prediction (b).

**Isolating the quality effect.** To quantify the incremental loss due to degraded quality at fixed $N$ and $D$, define

$$\Delta L(Q) = L(N, D, Q) - L(N, D, 1). \tag{4}$$

From the fitted law, this difference takes the form

$$\Delta L(Q) \approx \hat{B} \, D^{-\hat{\beta}} \left( Q^{-\hat{\gamma}} - 1 \right),$$

which makes clear that $\Delta L(Q)$ should grow approximately linearly in $Q^{-\hat{\gamma}} - 1$, where $\hat{\gamma}$ is the estimated $\gamma$. We therefore directly plot $\Delta L(Q)$ versus $Q^{-\hat{\gamma}} - 1$, using the estimated $\hat{\gamma}$ from our scaling experiments.

As shown in Figure 3, the resulting plots are close to linear across the tested quality levels, validating that the multiplicative $Q^{-\gamma}$ term accurately captures the degradation due to corruption. Furthermore, the plots show homogeneous linear functions, which suggest that the additive terms in our proposed scaling law, $\frac{A}{N^\alpha} + E$, indeed do not vary with data quality $Q$. The stability of these plots across dataset sizes (0.1B, 1B, 10B for CLM; 0.5M, 1M, 2M pairs for NMT) (with the Huber estimation) further suggests that $\gamma$ is an intrinsic task-dependent parameter rather than an artifact of optimization or dataset sampling.

In practice, the exponent $\gamma$ serves as a robustness index: smaller values indicate that a model-task pair is resilient to corruption, while larger values reveal greater sensitivity to data quality. In our

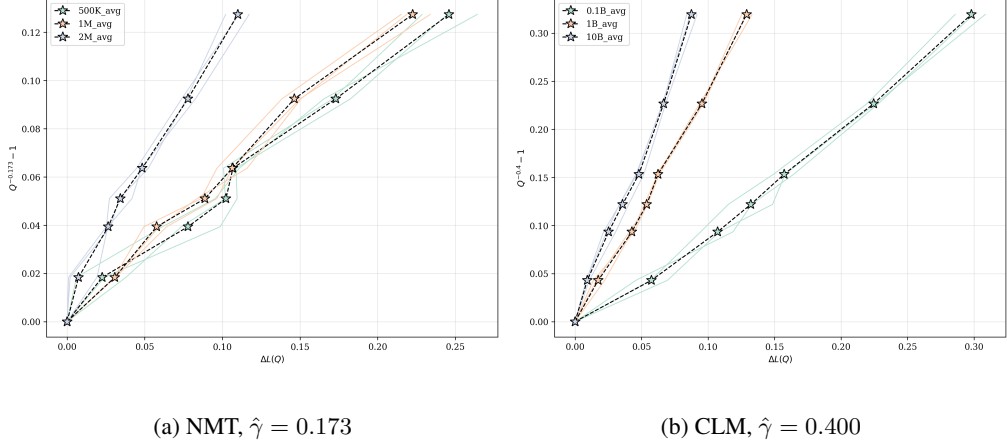

(a) NMT, $\hat{\gamma} = 0.173$        (b) CLM, $\hat{\gamma} = 0.400$

Figure 3: $\Delta L(Q)$ vs. $Q^{-\hat{\gamma}} - 1$ for a) NMT and b) CLM, where $\hat{\gamma}$ is the estimated $\gamma$ parameter using the Huber estimation from the scaling experiments.

experiments, both tasks show $\gamma < 1$, which means loss increases more slowly than linearly with decreasing $Q$. In other words, the model is robust to moderate amounts of corruption — one needs a large drop in $Q$ to see a noticeable rise in loss.

**Robustness and generalization.** We have included additional validations to verify the robustness of our findings (see Appendix E for full details). First, we confirmed that our synthetic noise model serves as a valid proxy for semantic degradation by observing a strictly monotonic decrease in embedding similarity with increasing noise (Figure 4). We performed learning rate sweeps and found that $\gamma$ is stable across practical learning rate regimes (Table 9). We provide a practical guide for estimating $Q$ in Section E.3.

## 6 SUMMARY AND CONCLUSION

This paper revisits scaling laws for large language model pretraining by introducing an explicit and dimensionless measure of data quality, alongside traditional factors like model size and dataset volume. We propose a new quality-aware scaling law that extends the widely used Chinchilla framework, predicting pretraining loss as a joint function of model size, data volume, and data quality. By systematically injecting synthetic noise in neural machine translation and causal language modeling experiments, we demonstrate that higher data quality leads to significantly lower loss for a given model size and dataset. Notably, our findings show that for high-quality datasets, smaller models and less compute are needed to achieve strong results, which is highly relevant for domain-specific applications. The study provides a formal scaling law:

$$L(N, D, Q) = \frac{A}{N^\alpha} + \frac{B}{D^\beta Q^\gamma} + E,$$

where $Q$ measures data quality and $\gamma$ is empirically estimated. This extends earlier work that primarily focused on model and data size. Our experiments on both NMT and CLM tasks, using real datasets with systematically controlled levels of data corruption, confirm the predictive power of the new scaling law and offer concrete guidance for balancing data curation versus model scale. The work establishes a unified framework that quantifies data quality's influence, enabling principled decisions for specialized LLM development in fields building smaller models over domain specific datasets, such as those that arise in business, scientific and medical applications.

### REPRODUCIBILITY STATEMENT

We include all results from our experimental runs, all model and training configurations and recipes and all steps taken to produce this work in the relevant sections in main and the appendix.

ACKNOWLEDGMENTS

This research was supported, in part, funded by the National Heart, Lung, and Blood Institute (NHLBI) of the National Institutes of Health (NIH). The views and conclusions contained in this document are those of the authors and should not be interpreted as representing the official policies, either expressed or implied, of the NIH/NHLBI. We also acknowledge the University of Chicago's Research Computing Center for their support of this work.

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

# A    PROOFS

## A.1    PROOF OF LEMMA 1

*Proof of Lemma 1.* For the Gaussian location family, the log-likelihood of a single observation is $\ell(\theta) = -\frac{1}{2\sigma^2}(y - \theta)^2 + \text{const}$, where $\theta = f(x)$. Differentiating twice with respect to $\theta$ and taking expectation over $\epsilon$ gives the Fisher information per sample:

$$\mathcal{I}_1(\theta) = \mathbb{E}\left[-\frac{\partial^2}{\partial \theta^2}\ell(\theta)\right] = \frac{1}{\sigma^2}.$$

By additivity of Fisher information across i.i.d. samples, $\mathcal{I}_D = D \cdot \mathcal{I}_1 = D/\sigma^2$. Thus, the usable information is equivalent to $D_{\text{eff}} = D \cdot (1/\sigma^2)$ noise-free samples. We define the deficiency as

$$\Delta = \ln\frac{\sigma^2}{\sigma_0^2}, \qquad Q = e^{-\Delta} = \frac{\sigma_0^2}{\sigma^2}.$$

where $\sigma_0^2$ is baseline noise (i.e. noise for "clean" data; wlog $\sigma_0 := 1$). In this formulation, datasets noisier than the reference ($\sigma^2 > \sigma_0^2$) have $Q < 1$, while "clean" datasets ($\sigma^2 = \sigma_0^2$) have $Q = 1$. The effective sample size then scales as

$$D_{\text{eff}} = D \cdot Q^\gamma$$

and taking $\gamma = 1$ recovers the classical $D \cdot \text{SNR}$ scaling.    $\square$

## A.2    PROOF OF LEMMA 2

*Proof of Lemma 2.* For symmetric label noise with flip rate $\eta$, the *unbiased loss correction* constructs a per-sample estimator $\tilde{\ell}(f(X), \tilde{Y})$ whose expectation matches the clean loss: $\mathbb{E}[\tilde{\ell} \mid X, Y] = \ell(f(X), Y)$. For symmetric noise, the correction has the form $\tilde{\ell} = \frac{(1-\eta)\,\ell(f(X),\tilde{Y}) - \eta\,\ell(f(X),1-\tilde{Y})}{1-2\eta}$ (Natarajan et al., 2013). This divides by $(1 - 2\eta)$, so the variance of $\tilde{\ell}$ inflates by a factor $(1 - 2\eta)^{-2}$ relative to the clean loss. Uniform convergence or stability-based generalization bounds then inherit a $\sqrt{\text{Var}[\tilde{\ell}]/D}$ dependence, implying an excess-risk rate equivalent to having $D_{\text{eff}} = D\,(1 - 2\eta)^2$ clean samples. Classic PAC analyses of RCN similarly show polynomial overheads governed by $(1 - 2\eta)^{-2}$ (e.g., Angluin and Laird (1988)).    $\square$

## A.3    PROOF OF PROPOSITION 1

*Proof of Proposition 1.* By assumption (as in the Information Bottleneck principle (Tishby et al., 1999)), the data-dependent loss scales as

$$L_D \propto \left(D \cdot I(\tilde{X}; Z)\right)^{-\beta}, \qquad \beta > 0.$$

Substituting the multiplicative reduction of usable information,

$$I(\tilde{X}; Z) = \rho(Q)\,I(X; Z),$$

we obtain

$$L_D \propto D^{-\beta}\,\rho(Q)^{-\beta}\,I(X; Z)^{-\beta}.$$

Since $I(X; Z)$ does not depend on $Q$ or $D$, we can absorb it into the constant $B$. By the regular-variation assumption, $\rho(Q) \approx c\,Q^{\gamma_0}$ as $Q \to 1$, so

$$\rho(Q)^{-\beta} \approx c^{-\beta}\,Q^{-\beta\gamma_0}.$$

Absorbing $c^{-\beta}$ into $B$ and setting $\gamma = \beta\gamma_0$, we obtain

$$L_D \approx \frac{B}{D^\beta Q^\gamma}.$$

Adding the standard model-size contribution $A/N^\alpha$ and irreducible error floor $E$ yields

$$L(N, D, Q) \approx \frac{A}{N^\alpha} + \frac{B}{D^\beta Q^\gamma} + E,$$

which is the claimed quality-aware scaling law after reparameterization of constants.    $\square$

## B    Extended Methods

### B.1    Tokenizer Training

We pretrain a BPE tokenizer for our experiments in Neural Machine translation from English to German. We use the HuggingFace Wolf et al. (2020) tokenizers module with the following pipeline. We use NFKC normalization with Whitespace pre-tokenization with a 32000 vocabulary size and a minimum frequency of 2 and $< \mathbf{unk} >$ , $< \mathbf{s} >$ , $< /\mathbf{s} >$ , $< \mathbf{pad} >$ , $< \mathbf{sep} >$ as the special tokens. Predictably $< \mathbf{s} >$ and $< /\mathbf{s} >$ are the bos and eos tokens respectively.$< \mathbf{sep} >$ is used as the separator token to separate the source and target languages.

### B.2    Model Configurations

We present the exact configurations that we use for our model definitions for each of the tasks. We use a GPT Neo model for NMT and a Llama model for CLM. Table 4 and Table 5 show the exact parameter definitions used for the two models. We use the official implementations exposed via HuggingFace transformers.

| Variable | Value |
|---|---|
| activation function | gelu new |
| attention dropout | 0.1 |
| attention types | [global, local] - 4 |
| classifier dropout | 0.1 |
| embed dropout | 0.1 |
| hidden size | 1024 |
| initializer range | 0.02 |
| layer norm epsilon | 1e-05 |
| max position embeddings | 260 |
| num heads | 8 |
| num layers | 8 |
| resid dropout | 0.1 |
| vocab size | 32000 |
| window size | 256 |

Table 4: GPT Neo config used for NMT pretraining experiments

### B.3    Data Pre-processing

#### B.3.1    Paracrawl v8

In preparing our dataset for machine translation from English to German, we used a pretrained language detection model from FastText (Joulin et al., 2016) lid.176.bin to filter the data as described in 5.1.

#### B.3.2    C4

We use a cleaned version of C4 by AllenAI Dodge et al. (2021) available on HuggingFace, they provide 5 splits, en, en.noclean, en.noblocklist, realnewslike, and multilingual (mC4). We use the en variant in our experiments, it contains 364,868,892 samples in the train split and 364,608 samples in the validation split.

## C    Empirical Results

### C.1    Additional Experimental Details

Table 6 shows other hyperparameters and choices made during our training experiments. We also notably use greedy dataset packing in NMT and make use of DataCollatorWithFlattening() from

| Variable | Value |
|---|---|
| attention bias | false |
| attention dropout | 0.0 |
| head dim | 64 |
| hidden act | silu |
| hidden size | 512 |
| initializer range | 0.02 |
| intermediate size | 2048 |
| max position embeddings | 2048 |
| mlp bias | false |
| model type | llama |
| num attention heads | 8 |
| num hidden layers | 8 |
| num key value heads | 8 |
| pretraining tp | 1 |
| rms norm eps | 1e-05 |
| rope scaling | null |
| rope theta | 10000.0 |
| tie word embeddings | true |
| vocab size | 128256 |

Table 5: Llama config used for CLM pretraining experiments

HuggingFace in CLM to train to easily use Flash Attention 2 (Dao, 2023). Additionally we use weight decay of 0.01 in both cases.

Table 6: Training Details

| TASK | D(avg) | Batch Size | Max LR | Grad Acc Steps | Warmup Ratio |
|---|---|---|---|---|---|
| NMT | 3.6754244e+07 | 512 | 5e-4 | 8 | 0.2 |
| NMT | 7.3486728e+07 | 512 | 5e-4 | 8 | 0.2 |
| NMT | 1.4697256e+08 | 1024 | 5e-4 | 8 | 0.2 |
| CLM | 1.0317308e+08 | 48 | 1e-3 | 1 | 0.1 |
| CLM | 1.0296028e+09 | 48 | 1e-3 | 8 | 0.1 |
| CLM | 1.0294779e+10 | 48 | 1e-3 | 8 | 0.1 |

## C.2 ADDITIONAL RESULTS

In reporting our performance we evaluate our trained models on test loss on heldout datasets. In NMT, during dataset prepartion we partition a test set of 50,000 samples and report evaluation loss on this heldout. For CLM, we partition 40,000 samples from the validation set of C4 en and use that for evaluation. The full results are shown in Table 7 and Table 8 for reference.

## C.3 PARAMETRIC LOSS FITTING WITH HUBER LOSS

Similar to Hoffmann et al. (2022) we minimize the following objective following a regularization introduced by Huber (1992)

$$\min_{b,e,\beta,\gamma} \sum_{\text{Run } i} \text{Huber}_\delta \Big( \text{LSE}\big(b - \beta \log D_i - \gamma \log Q_i, e\big) - \log L_i \Big)$$

where $B = \exp(b)$, $E = \exp(e)$ and $\delta = 10^{-3}$ using the L-BFGS-B solver available through SciPy minimize. We replicate the grid initialization of parameters as follows, $b \in [0, 25, 5]$, $e \in [0, 2, 0.5]$, $\beta \in [0.0, 0.4, 0.1]$ and $\gamma \in [0.0, 0.4, 0.1]$. We also add bounds $0 \le \beta \le 1$ and $0 \le \gamma \le 1$. We also report a secondary fit using SciPy inbuilt function curve fitting functions using Least Squares.

Table 7: Neural Machine Translation Results. $D$ is the number of tokens used for training , $Q$ represents dataset quality and $L$ is the test loss in nats. The horizontal lines demarcate the replicate experiments and vertical sections the different dataset sizes used.

| D | Q | L | D | Q | L | D | Q | L |
|---|---|---|---|---|---|---|---|---|
| 36779001 | 1.00 | 1.853 | 73479862 | 1.00 | 1.565 | 146952084 | 1.00 | 1.351 |
| 36779001 | 0.90 | 1.859 | 73479862 | 0.90 | 1.596 | 146952084 | 0.90 | 1.370 |
| 36779001 | 0.80 | 1.914 | 73479862 | 0.80 | 1.629 | 146952084 | 0.80 | 1.377 |
| 36779001 | 0.75 | 1.949 | 73479862 | 0.75 | 1.662 | 146952084 | 0.75 | 1.393 |
| 36779001 | 0.70 | 1.964 | 73479862 | 0.70 | 1.672 | 146952084 | 0.70 | 1.399 |
| 36779001 | 0.60 | 2.018 | 73479862 | 0.60 | 1.715 | 146952084 | 0.60 | 1.434 |
| 36779001 | 0.50 | 2.117 | 73479862 | 0.50 | 1.799 | 146952084 | 0.50 | 1.468 |
| 36734209 | 1.00 | 1.842 | 73487922 | 1.00 | 1.572 | 147005794 | 1.00 | 1.358 |
| 36734209 | 0.90 | 1.866 | 73487922 | 0.90 | 1.602 | 147005794 | 0.90 | 1.359 |
| 36734209 | 0.80 | 1.941 | 73487922 | 0.80 | 1.631 | 147005794 | 0.80 | 1.384 |
| 36734209 | 0.75 | 1.952 | 73487922 | 0.75 | 1.655 | 147005794 | 0.75 | 1.386 |
| 36734209 | 0.70 | 1.951 | 73487922 | 0.70 | 1.688 | 147005794 | 0.70 | 1.403 |
| 36734209 | 0.60 | 2.025 | 73487922 | 0.60 | 1.722 | 147005794 | 0.60 | 1.432 |
| 36734209 | 0.50 | 2.087 | 73487922 | 0.50 | 1.791 | 147005794 | 0.50 | 1.469 |
| 36749529 | 1.00 | 1.853 | 73492417 | 1.00 | 1.576 | 146959769 | 1.00 | 1.360 |
| 36749529 | 0.90 | 1.891 | 73492417 | 0.90 | 1.607 | 146959769 | 0.90 | 1.362 |
| 36749529 | 0.80 | 1.927 | 73492417 | 0.80 | 1.626 | 146959769 | 0.80 | 1.388 |
| 36749529 | 0.75 | 1.955 | 73492417 | 0.75 | 1.662 | 146959769 | 0.75 | 1.394 |
| 36749529 | 0.70 | 1.954 | 73492417 | 0.70 | 1.673 | 146959769 | 0.70 | 1.412 |
| 36749529 | 0.60 | 2.024 | 73492417 | 0.60 | 1.715 | 146959769 | 0.60 | 1.437 |
| 36749529 | 0.50 | 2.082 | 73492417 | 0.50 | 1.791 | 146959769 | 0.50 | 1.462 |

Table 8: Causal Language Modeling Results. $D$ is the number of tokens used for training , $Q$ represents dataset quality and $L$ is the test loss in nats. The horizontal lines demarcate the replicate experiments and vertical sections the different dataset sizes used.

| D | Q | L | D | Q | L | D | Q | L |
|---|---|---|---|---|---|---|---|---|
| 103068758 | 1.00 | 4.401 | 1029888156 | 1.00 | 3.824 | 10295030326 | 1.00 | 3.581 |
| 103068758 | 0.90 | 4.447 | 1029888156 | 0.90 | 3.846 | 10295030326 | 0.90 | 3.592 |
| 103068758 | 0.80 | 4.520 | 1029888156 | 0.80 | 3.867 | 10295030326 | 0.80 | 3.612 |
| 103068758 | 0.75 | 4.534 | 1029888156 | 0.75 | 3.876 | 10295030326 | 0.75 | 3.621 |
| 103068758 | 0.70 | 4.566 | 1029888156 | 0.70 | 3.886 | 10295030326 | 0.70 | 3.633 |
| 103068758 | 0.60 | 4.630 | 1029888156 | 0.60 | 3.917 | 10295030326 | 0.60 | 3.649 |
| 103068758 | 0.50 | 4.701 | 1029888156 | 0.50 | 3.957 | 10295030326 | 0.50 | 3.673 |
| 103036697 | 1.00 | 4.402 | 1029558108 | 1.00 | 3.825 | 10292963774 | 1.00 | 3.584 |
| 103036697 | 0.90 | 4.472 | 1029558108 | 0.90 | 3.841 | 10292963774 | 0.90 | 3.592 |
| 103036697 | 0.80 | 4.510 | 1029558108 | 0.80 | 3.866 | 10292963774 | 0.80 | 3.607 |
| 103036697 | 0.75 | 4.550 | 1029558108 | 0.75 | 3.880 | 10292963774 | 0.75 | 3.617 |
| 103036697 | 0.70 | 4.560 | 1029558108 | 0.70 | 3.886 | 10292963774 | 0.70 | 3.629 |
| 103036697 | 0.60 | 4.628 | 1029558108 | 0.60 | 3.921 | 10292963774 | 0.60 | 3.650 |
| 103036697 | 0.50 | 4.710 | 1029558108 | 0.50 | 3.950 | 10292963774 | 0.50 | 3.668 |
| 103413788 | 1.00 | 4.407 | 1029362179 | 1.00 | 3.824 | 10296342093 | 1.00 | 3.581 |
| 103413788 | 0.90 | 4.464 | 1029362179 | 0.90 | 3.837 | 10296342093 | 0.90 | 3.589 |
| 103413788 | 0.80 | 4.502 | 1029362179 | 0.80 | 3.868 | 10296342093 | 0.80 | 3.602 |
| 103413788 | 0.75 | 4.523 | 1029362179 | 0.75 | 3.878 | 10296342093 | 0.75 | 3.614 |
| 103413788 | 0.70 | 4.557 | 1029362179 | 0.70 | 3.887 | 10296342093 | 0.70 | 3.627 |
| 103413788 | 0.60 | 4.627 | 1029362179 | 0.60 | 3.920 | 10296342093 | 0.60 | 3.646 |
| 103413788 | 0.50 | 4.693 | 1029362179 | 0.50 | 3.952 | 10296342093 | 0.50 | 3.667 |

# D  RELATIONSHIP TO OTHER DEFINITIONS OF DATA QUALITY

In this section, we show that our quality aware scaling law recovers the scaling law in Chen et al. (2025) by assuming that the data deficiency $\Delta(\omega)$ for a dataset $\omega$ has the following form:

$$\Delta(\omega) = \mu_1 E + \mu_2 \frac{1}{F} + \mu_3 G \tag{5}$$

where $E$ is the total noise, $F$ is the coverage scaled between 0 (minimum coverage) and 1 (maximum coverage), and $G$ quantifies the amount of repeated data.

We assume that density as defined in Chen et al. (2025), which incorporates both coverage and repeated data, is quantified such that low density is associated with high quality data and high density is associated with lower quality data.

Let $\rho$ be the density of a dataset as defined in Chen et al. (2025), Equation 3, with the density assumed to satisfy $0 \leq \rho \leq 1$. Here low $\rho$ close to 0 corresponds to higher quality data with fewer repetitions, and high $\rho$ close to 1 corresponds to low quality data with more repetitions, leading to sub-scaling behavior Chen et al. (2025).

We define the data dispersion to be

$$\text{Dis}(\rho) = -\ln(1 - \rho),$$

so as $\rho$ approaches 1 (low quality data in the sense of Chen et al. (2025)), Dis approaches $\infty$, and as $\rho$ approaches 0 (high quality data in the sense of Chen et al. (2025)), Dis approaches 1.

In the equation above, we assume that the data dispersion Dis captures the coverage $1/F$ of the data and density $\rho$. We also assume that the Dis captures the amount of repeated data, although as we will see below, this can also be modeled in more precise ways. If we fix the noise $E$, then the data deficiency is

$$\Delta(\omega) = \text{constant} + \mu_2 \text{Dis}(\rho) \tag{6}$$

and we recover the data quality scaling law in Chen et al. (2025). In particular, as the density $\rho$ approaches 0, the data quality factor $Q$ approaches 1, and as the density $\rho$ approaches 1, the data quality degrades.

More generally, we can define a modified data dispersion

$$\text{Dis}^\kappa(\rho) = -\ln(1 - \rho^\kappa).$$

Turning towards the data quality scaling law in Chang et al. (2024), if instead of using the cluster based definition of density, we use the compression based definition of density, we get

$$\text{CR}(\omega) = \text{size}(\omega)/\text{compressed size}(\omega)$$
$$\text{DR}(\omega) = \text{CR}^{-1}(\omega)$$

We define dispersion as:

$$\text{Dis}(\rho) = -\ln(1 - \text{DR}(\omega)).$$

With this definition of $\text{Dis}(\omega)$, the rest of the argument is the same.

In a similar fashion, we can add a term $H = S(\omega)$ to the data deficiency $\Delta(\omega)$ to capture the increase in data deficiency due to the negative impact of synthetic data, which we can quantify with a term as follows

$$\Delta(\omega) = \mu_1 E + \mu_2 \frac{1}{F} + \mu_3 G + \mu_4 H$$

For example, Chang et al. (2024) measures $S(\omega)$ using the perplexity measure.

Turning to Goyal et al. (2024), we can define a notion of the declining value of data as the number of training epochs $k$ increases (and hence the amount of repeated data increases). We can define an additive term in the data deficiency as follows:

$$\mu_3 \frac{k}{\tau}$$

where $k$ is the number of epochs. With this form, for small number of epochs, the loss decreases, but as the number of epochs increases, the loss begins to grow exponentially, and then levels out, with the rate determined by a "decay parameter" $\tau$.

## E    ADDITIONAL VALIDATIONS AND PRACTICAL GUIDE

In this section, we present additional experiments validating our noise model and scaling law stability, along with a practical guide for estimating $Q$.

### E.1    VALIDATION OF SYNTHETIC NOISE MODEL

To validate that our synthetic noise injection serves as a meaningful proxy for real-world data degradation, we measured the semantic drift of the corrupted data. We computed the average cosine similarity between the embeddings of clean samples and their synthetically corrupted counterparts using a pre-trained reference model. As shown in Figure 4 there is a strictly monotonic, near-linear increase in embedding similarity as the synthetic quality increase (noise level decrease). This confirms that our parameter $Q$ effectively captures the loss of semantic information.

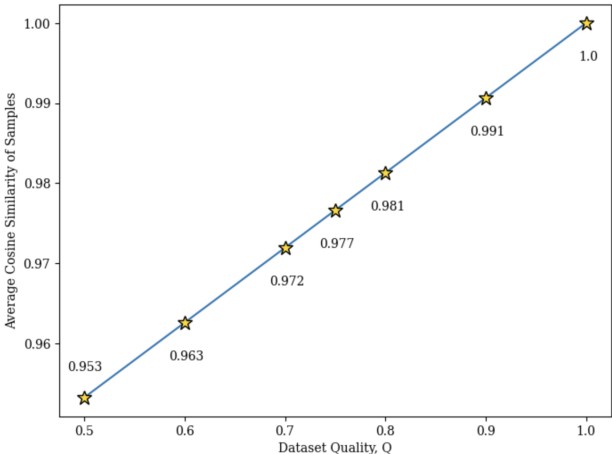

Figure 4: Semantic drift analysis: Average cosine similarity between embeddings of clean and corrupted samples at various quality levels.

### E.2    SENSITIVITY TO LEARNING RATE

We performed a sensitivity analysis by retraining models at 5 different learning rates (1e-5 to 5e-3) across 3 dataset scales and 3 noise levels. As shown in Table 9 and Figure 5, the estimated $\gamma$ is highly stable ($\sim 0.39$) for practical learning rates near the optimal 1e-3 used in our main experiments.

### E.3    PRACTITIONER'S GUIDE TO ESTIMATING Q

We propose the following recipe for estimating $Q$ in practice:

1. **Establish a baseline:** Identify a small, high-quality reference subset (e.g., manually curated text or a trusted domain corpus) to represent $Q \approx 1$.
2. **Select quality proxies:** Choose measurable proxies for "deficiency" relevant to the domain.

Table 9: Sensitivity of the estimated quality exponent $\gamma$ to learning rate.

| Learning Rate | Estimated $\gamma$ |
|---|---|
| 1e-5 | 0.037 (Undertrained) |
| 3e-4 | 0.394 |
| **1e-3 (Ours)** | **0.396** |
| 5e-3 | 0.177 (Unstable) |

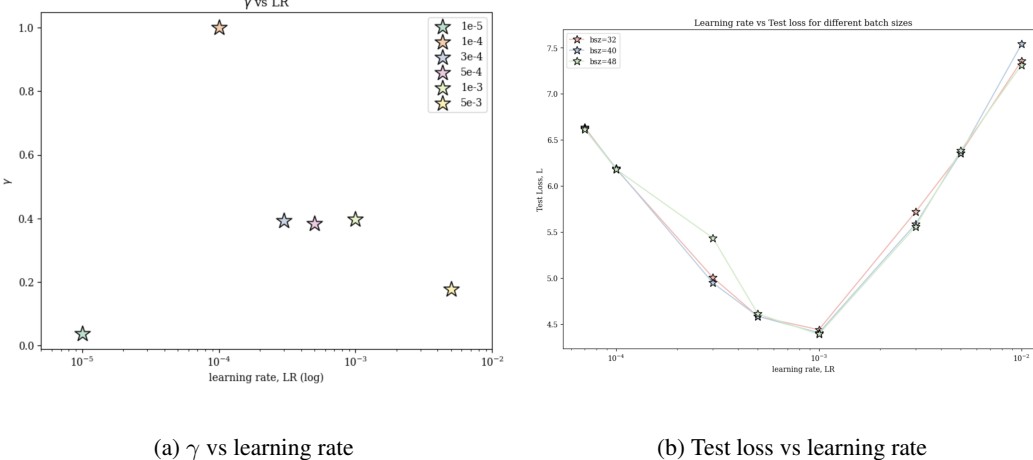

(a) $\gamma$ vs learning rate        (b) Test loss vs learning rate

Figure 5: Sensitivity analysis of learning rate: (a) Estimated $\gamma$ across different learning rates, (b) Learning rate vs test loss for different batch sizes.

3. **Compute deficiency ($\Delta$):** Construct a composite deficiency score $\Delta = \sum_j \mu_j \Delta_j$, where $\mu_j$ are weights determined by importance or preliminary ablation.

4. **Map to $Q$:** Calculate $Q = e^{-\Delta}$. This maps the unbounded deficiency metric to the $(0, 1]$ interval.

5. **Sanity check:** Verify that $Q$ correlates with downstream performance or semantic drift (as in our embedding experiment) on a small validation set.

6. **Fit scaling law:** Use the estimated $Q$ values for different data subsets to fit the parameters $(B, \beta, \gamma)$ of the scaling law $L \approx B/(D^\beta Q^\gamma) + E$ (assuming $N$ is fixed or large enough), using the robust Huber loss method described in the paper.

