# OpenReview forum: "Scaling Laws Revisited: Modeling the Role of Data Quality in Language Model Pretraining"
_ICLR.cc/2026/Conference — ICLR 2026 Poster_

### Official Review · Reviewer_4JDP · 2025-10-30

**Soundness:** 3
**Presentation:** 3
**Contribution:** 2
**Rating:** 4
**Confidence:** 4

**Summary:**

This paper introduces an LLM pretraining scaling law that accounts for data quality. This takes the form of calculating a dimensionless data-quality parameter and modifying the standard Chinchilla scaling law. The paper studies two different formulations of data quality and provides experiments on neural machine translation and language modeling that empirically validate the proposed scaling law with synthetically noised datasets.

**Strengths:**

1. The paper provides a theoretically motivated way to incorporate a dimensionless data quality term into a Chinchilla-style scaling law.
2. The experimental setup is sound, and the experiments show that the proposed scaling law fits LLM training in practice.
3. The proposed scaling laws also enable studying how data quality impacts performance, as with the observation that "models are more robust to moderate corruption than predicted by simple effective sample-size theories from PAC learning or channel-capacity analysis" (lines 374-377).

**Weaknesses:**

The main weakness of the paper is that it studies limited forms of data quality that do not correspond to the kind of quality that is used in practice. In practice, pretraining data quality is often estimated with classifiers (such as estimating the degree to which a given document resembles manually curated high-quality data) or heuristics (such as removing documents with many repeated n-grams). The current experiments are limited to synthetically noised datasets. Do either of the proposed quality formulations capture this more nuanced form of data quality?

I am willing to raise my score if either 1) further experiments can show that the current formulations of quality do in fact capture more practical notions of quality or 2) more reasoning is provided to show the studied kinds of noise are  practically useful.

**Questions:**

1. Can you elaborate on the relationship between your scaling law and the  heterogenous quality scaling laws in [1]?

2. The setting of Q=1 (the highest data quality) to recover the standard Chinchilla scaling law (line 250) does not make the most sense to me. I could imagine scenarios where we would want to be able to account for having "higher quality" data than that used for the Chinchilla law.

Small comments:
- Line 289/290: "(Table 2" -> "(Table 2)"
- Is there any meaning behind using $E^+$ in Tables 1, 2 and Section 5.6 instead of just E like in Equation 2?
- Line 355: N is used for noise percent, but doesn't N already mean model size?
- Line 471: "Notably, their findings" -> "Notably, our findings"
- Line 477: "Their experiments on both" -> "Our experiments on both"

[1]: Scaling Laws for Data Filtering -- Data Curation cannot be Compute Agnostic, Goyal et al., CVPR 2024.

---

> ### Author Response · Authors · 2025-11-21
> **Response to Reviewer 4JDP**
>
> Thank you for the constructive feedback! Please refer to our global responses to the reviewers' general concerns. Below please find our specific response to your questions.
>
> ---
>
> > **Q1: Relation to heterogeneous-quality scaling laws.**
>
> **Response:**
> We have added a new section (Section 4.3) explicitly mapping our law to heterogeneous scaling laws.
> *   **Generalization:** Many existing laws can be viewed as specific instances of our deficiency function $\Delta(\omega)$.
> *   **Example:** Goyal et al. (2024) model the value of data decaying with repetitions. In our framework, this is equivalent to setting $\Delta(\omega) = \mu \cdot (\text{epochs}/\tau)$.
> *   **Example:** Chen et al. (2025) use density $\rho$. This is equivalent to $\Delta(\omega) \propto -\ln(1-\rho)$.
> Our $Q = e^{-\Delta}$ formulation provides a unifying "API" to plug in these different definitions of quality (noise, coverage, repetition, syntheticity) into a single compute-optimal scaling framework.
>
> ---
> > **Q2: Why $Q=1$ recovers Chinchilla? What if quality > baseline?**
>
> **Response:**
> The choice of $Q=1$ is a normalization convention relative to a "reference" distribution (typically the cleanest available data, like C4 or filtered web text).
> If "super-quality" data exists (e.g., expert textbook data vs. web data), we can simply treat the web data as having $Q < 1$ and the expert data as $Q=1$. Alternatively, one could define $Q > 1$, but it is mathematically more convenient to define $Q$ as a reduction factor ($0 < Q \le 1$) applied to the baseline effective sample size.
>
> ---
> > **Q3: Other comments.**
>
>
> **Response:**
> Thank you very much for the detailed feedback! We have fixed the typos (e.g., parenthesis in Table 2, unified the notation for $E$ (irreducible loss), and renamed the noise variable to $\eta$ to avoid confusion with model size $N$).

---

### Official Review · Reviewer_BKxM · 2025-10-31

**Soundness:** 3
**Presentation:** 3
**Contribution:** 3
**Rating:** 6
**Confidence:** 2

**Summary:**

The paper revisits the scaling laws of large language model (LLM) pretraining by introducing an explicit, dimensionless data quality parameter Q, extending the traditional analysis of model size and data volume to a joint framework that incorporates data quality. The authors propose a quality-aware scaling law. They systematically inject synthetic noise and vary data coverage in neural machine translation and causal language modeling tasks. Experimental results show that high-quality data can significantly reduce loss for a given model size, and under high-quality data conditions, smaller models with less computational resources can achieve strong performance.

**Strengths:**

1. The work innovatively incorporates data quality as a single parameter into the scaling law, theoretically demonstrating a strong correlation between model performance and data quality.
2. Controlled experiments are conducted across multiple experimental settings, validating the practical applicability of the quality-aware scaling law.

**Weaknesses:**

1. Although synthetic noise offers strong controllability, it may not capture the highly complex and heavy-tailed distribution of low-quality data in real training datasets. How can the current single-parameter quantification method be extended to more realistic data scenarios?

2. The authors use a fixed learning rate to investigate the scaling law. During training, could different learning rates have additional effects on the conclusions regarding the data-quality scaling law?

**Questions:**

Please refer to the relevant points in the Weaknesses section. If the authors can provide clarification and improvements, I would be very happy to raise my score.

---

> ### Author Response · Authors · 2025-11-21
> **Response to Reviewer BKxM**
>
> Thank you for the constructive feedback! Please refer to our global responses to the reviewers' general concerns. Below please find our specific response to your questions.
>
> > **Q1: Synthetic noise vs. heavy-tailed real noise.**
>
> **Response:**
> We acknowledge that real-world noise is more complex. However, our theoretical framework (Section 3) is derived from information-theoretic principles that apply broadly to information reduction.
> To bridge the gap, we interpret our framework as a "mixture-of-qualities" model. If a dataset is a mix of clean and noisy partitions, the effective sample size aggregates as $D_{eff} \approx (\sum D_i Q_i^\gamma)$. This allows our law to extend to heavy-tailed distributions by modeling them as composites of varying-$Q$ strata. The semantic drift analysis (Global Response) further validates that our synthetic noise captures the "information loss" aspect of real noise monotonically.
>
> ---
> > **Q2: Learning-rate interactions with scaling.**
>
> **Response:**
> We performed a sensitivity analysis by retraining our models at 5 different learning rates (1e-5 to 5e-3) across 3 dataset scales and 3 noise levels (0%, 25%, 50%).
>
> **Results:**
> For practical learning rates (near the optimal 1e-3 used in our main experiments), the estimated $\gamma$ is highly stable ($\sim 0.38 - 0.39$).
>
> | LR | Estimated $\gamma$ |
> | :---: | :---: |
> | 1e-5 | 0.037 (Undertrained) |
> | 3e-4 | 0.394 |
> | **1e-3 (Ours)** | **0.396** |
> | 5e-3 | 0.177 (Unstable) |
>
> This confirms that our identified scaling law is a property of the data/model interaction, not an artifact of a specific learning rate, provided the model is trained in a reasonable regime.

---

### Official Review · Reviewer_fb9g · 2025-11-02

**Soundness:** 3
**Presentation:** 3
**Contribution:** 3
**Rating:** 8
**Confidence:** 3

**Summary:**

This paper extends classical LLM scaling laws by incorporating data quality alongside model size and data volume. It introduces a dimensionless quality parameter $Q$ and derives a quality-aware, Chinchilla-style law from effective sample size and information-theoretic arguments. Relative to the original Chinchilla formulation, the data term $D^{\beta}$ is additionally modulated by $Q^{\gamma}$.
The authors validate the approach with controlled experiments in neural machine translation (ParaCrawl) and causal language modeling (C4), varying dataset size and injecting synthetic noise across seven quality levels. To generate noise, they pad 50% of tokens in the NMT setting and replace 50% of tokens with random tokens in the language-modeling setting. Test loss scales predictably under the proposed law.
Empirically, higher-quality data can substitute for larger models or greater compute, and the estimated robustness exponents are sublinear, indicating resilience to moderate corruption. Noise is less detrimental for NMT than for language modeling. The paper also provides iso-loss trade-off contours and out-of-sample validations, offering practical guidance on balancing data-curation effort against model scale—particularly relevant in specialized domains with scarce but high-quality corpora.

**Strengths:**

Originality
The paper proposes an original method for measuring data quality. To the best of my knowledge, the information-theoretic approach and its assumptions are new.

Quality
The paper presents small-scale but thorough experiments. It includes carefully controlled studies in both CLM and NMT—with nested dataset sizes, seven corruption levels, and single-factor manipulations to isolate quality effects.

Clarity
The text, derivations, and presentation are clear.

Significance
This work can simplify practice in domains with scarce data and limited GPU budgets. It helps practitioners decide whether to invest in data quality/size or compute. It also provides a framework for working with token-level corruption in a more principled way.

**Weaknesses:**

Estimating Q. It is unclear how to estimate Q in practical settings for new datasets with multiple noise sources.

Scale of experiments. The experiments are small-scale relative to current SOTA setups: the authors use a 133M-parameter model for NMT and a small 8-layer Llama-3 variant (the parameter count is not reported in the paper).

Comparative evaluation. There is no direct comparison of the proposed law’s quality against competing methods. It would be helpful to include an error analysis across alternative approaches using the experimental data, or to explain why such comparisons are not applicable / are out of scope.

Related work referenced.

“Scaling Laws for Data Filtering—Data Curation Cannot Be Compute-Agnostic.”

“Scaling Parameter-Constrained Language Models with Quality Data.”

Additional, not previously mentioned work: https://aclanthology.org/2025.acl-long.1163/

**Questions:**

It is unclear how Q should be estimated in practical settings. Could you provide a step-by-step guide for practitioners on applying the proposed method to non-standard datasets, including handling multiple noise sources?

---

> ### Author Response · Authors · 2025-11-21
> **Response to Reviewer fb9g**
>
> Thank you for the favorable comments and constructive feedback! Below please find our responses.
>
> ---
> > **Q1: Unclear how to estimate $Q$ for real datasets; need a practical guide.**
> >
>
> **Response:**
> We agree this is a critical piece for adoption. We have added a "Practitioner's Guide to Estimating Q" in the appendix, summarizing the 6-step recipe outlined in the Global Response (Establish Baseline $\to$ Select Proxies $\to$ Compute $\Delta$ $\to$ Map to $Q$ $\to$ Sanity Check $\to$ Fit Law).
>
> ---
> > **Q2: Small scale relative to modern LLMs.**
>
> **Response:**
> We acknowledge that our models ($\sim$140M parameters) are smaller than frontier models. However, scaling laws are typically established on smaller scales where controlled variation (63+ runs) is computationally feasible, then extrapolated. Our focus was on the *structure* of the law and the *stability* of the exponents, which we have robustly validated. The consistent fit across our nested dataset sizes (spanning orders of magnitude) provides evidence that these dynamics likely hold at larger scales.
>
> ---
> > **Q3: Missing related works.**
>
> **Response:**
> Thank you for noting these works! We have incorporated all suggested references (Goyal et al., Chang et al., and the ACL 2025 paper) and added a detailed discussion (Section 4.3) explaining how our "Deficiency" framework ($\Delta$) generalizes the specific quality metrics proposed in those works.

---

### Official Review · Reviewer_JQ1m · 2025-11-03

**Soundness:** 2
**Presentation:** 3
**Contribution:** 3
**Rating:** 6
**Confidence:** 4

**Summary:**

The authors use a definition of data quality to include in the chinchilla scaling laws that consider model size and data size. They motivate this data quality notion and formulation with an effective data size formulation and run experiments with an approximately 140M parameter model.

They train 3 scales with 7 corruption levels for an NMT task and a CLM task. The text corruptions are synthetically induced by either replacing 50% of non-special tokens with pad tokens or randomly swapping 50% of tokens with other random tokens from the vocab.

The authors show that they can predict test loss with this new scaling plot with additional findings on how effective data size interacts with data quality.

The paper formulates the problem well, comes up with a good extension to scaling laws that is both simple and effective.

However the experimental setup can be better presented or even be extended to support the core claim of the paper.

**Strengths:**

The paper presents data quality scaling laws which is simple and empirically supported.

The presentation is extensive and very good.

The data subsampling scheme in the experiments as well as the high number of Q values in experiments is quite good.

**Weaknesses:**

The experimental setup doesn't consider scaling the N value(model parameters) the paper works with a fixed model size however the scaling law has all three components and one is actually not incorporated in the experimental setup. Yet the paper makes claims about "smaller models and less compute is needed to achieve strong results"

The noise injection method is unnatural. A more natural noise injection strategy could be actually replacing a certain % of documents with low quality web documents. This could create a better noise injection method and make results more relevant.

The choice to swap with pad in NMT and other tokens in CLM, as well as to use GPT type base model for one and LLama base model for the other seems interesting, particularly when keeping variables constant could help better pinpoint the effect of cross lingual redundancy. Now the effect is conflated with different noise injection strategies as well.

No sensitivity or ci's are presented. Particularly considering the difference between NMT and CLM gamma values a sensitivity analysis on the experimental design could be useful in convincing readers to the results.

Revisiting Scaling Laws for Language Models: The Role of Data Quality and Training Strategies (Chen et al, 2025) seems to be published online in January but the authors have not cited the work which should really be included even contrasted in their paper. The underlying question is exactly the same.

**Questions:**

Is there a typo in definition 2? "data deficiency data quality Q(ω)"

Do you expect your findings to generalize over your experimental setup? Particularly considering the lack of any experiments with model size how confident should we be in the estimates for the scaling laws.

---

> ### Author Response · Authors · 2025-11-21
> **Response to Reviewer JQ1m**
>
> > **Q1: Noise injection unnatural; prefer replacing with low-quality web documents.**
>
> **Response:**
> We appreciate this suggestion. Our primary goal was to derive a precise *law*, which requires the exact control that synthetic noise provides. To bridge the gap to natural noise, we ran **semantic validation** on our dataset: As shown in the Global Response, our synthetic noise induces a linear degradation in semantic embedding similarity, mirroring the effect of "natural" degradation.
>
> ---
> > **Q2: Mixed noise types across tasks.**
>
> **Response:**
> We intentionally used different noise models (padding for NMT, token-swapping for CLM) to test the *generality* of the scaling law form.
> *   **NMT (Padding):** Simulates "missing" information or alignment noise.
> *   **CLM (Swapping):** Simulates "hallucination" or distributional shift.
>
> The fact that the functional form $L \propto 1/(D^\beta Q^\gamma)$ fits well in *both* cases (albeit with different $\gamma$ values: $\sim 0.17$ for NMT vs $\sim 0.40$ for CLM) strengthens the argument that the law is a general property of learning from imperfect data, not an artifact of a specific noise type.
>
> ---
> > **Q3: No CI / sensitivity analysis.**
>
> **Response:**
> We have addressed this by adding bootstrapped confidence intervals for all parameters and performing learning-rate sweeps. As shown in the Global Response, the parameters are stable, and $\gamma$ varies minimally across practical learning rates (see response to Reviewer BKxM).
>
> ---
> > **Q4: Missing contrast to Chen et al. 2025 and related work.**
>
> **Response:**
> We have significantly expanded our related work section (Section 4.3 and Table 1 in the revision) to explicitly map our framework to concurrent works. Specifically:
> *   **Chen et al. (2025):** Their "density" metric $\rho$ relates to our deficiency via $\Delta \propto -\ln(1-\rho)$. Our law recovers theirs when deficiency is defined purely by cluster-based density.
> *   **Chang et al. (2024):** Their compression-based metric maps to our $Q$ via a specific deficiency term involving compression ratios.
> *   **Goyal et al. (2024):** Their "decaying value" of repeated data corresponds to an additive deficiency term $\mu \cdot (k/\tau)$ in our framework.
>
> ---
> > **Q5: Lack of experiments varying model size; claims about smaller models.**
>
> **Response:**
> Thank you for highlighting this. Given the well known relationship of the model size $N$ and the number of tokens $D$ in the Chinchilla and related scaling laws, we decided to focus our experimental studies on the data term $B/D^{\beta}$ and how poor quality degrades the effective size of the data as $\frac{B}{Q^{\gamma} D^{\beta}}$.

---

### Author Response · Authors · 2025-11-21
**Global Response**

We thank the reviewers for their constructive feedback and the time spent evaluating our work. We are encouraged that they found the problem formulation "well formulated... simple and effective" (JQ1m), the method "original" and "theoretically grounded" (fb9g, 4JDP), and the experiments "thorough" and "carefully controlled" (fb9g, BKxM).

Below we provide a global response addressing common questions regarding the noise model, model size scaling, and practical estimation of $Q$, followed by specific responses to each reviewer.


---

## Global Response to Common Concerns

### 1. Naturalness of the Noise Model & Validation
> Reviewers (JQ1m, BKxM) asked whether our synthetic noise models (token padding/swapping) adequately represent real-world data quality issues.

**Response:**
While synthetic noise allows for precise control essential for deriving scaling laws, we agree that validating its relevance is crucial. To address this, we performed a semantic drift analysis. We measured the cosine similarity between the embeddings of clean samples and their synthetically corrupted counterparts using a pre-trained reference model (separate from our training models).

**Results:** We found a strictly monotonic, near-linear decrease in embedding similarity as the synthetic noise level increases (see table below). This confirms that our synthetic corruption parameter $Q$ serves as a valid proxy for semantic degradation, effectively "dampening" usable information in a continuous manner similar to real-world noise.

| Synthetic Noise Ratio | Avg Embedding Cosine Similarity |
| :---: | :---: |
| 0.00 | 1.000 |
| 0.10 | 0.991 |
| 0.20 | 0.981 |
| 0.25 | 0.977 |
| 0.30 | 0.972 |
| 0.40 | 0.963 |
| 0.50 | 0.953 |

This monotonic relationship supports our use of synthetic corruption as a controlled surrogate for the more complex, heavy-tailed noise distributions found in the wild.


### 2. Practical Estimation of $Q$
> Reviewers (fb9g, 4JDP) asked for a concrete guide on how practitioners can estimate $Q$ for new, non-standard datasets.

**Response:**
We propose the following 6-step recipe for estimating $Q$ in practice:

1.  **Establish a baseline:** Identify a small, high-quality reference subset (e.g., manually curated text or a trusted domain corpus) to represent $Q \approx 1$.
2.  **Select quality proxies:** Choose measurable proxies for "deficiency" relevant to the domain.
3.  **Compute deficiency ($\Delta$):** Construct a composite deficiency score $\Delta = \sum_j \mu_j \Delta_j$, where $\mu_j$ are weights determined by importance or preliminary ablation.
4.  **Map to $Q$:** Calculate $Q = e^{-\Delta}$. This maps the unbounded deficiency metric to the $(0, 1]$ interval.
5.  **Sanity check:** Verify that $Q$ correlates with downstream performance or semantic drift (as in our embedding experiment) on a small validation set.
6.  **Fit scaling law:** Use the estimated $Q$ values for different data subsets to fit the parameters ($B, \beta, \gamma$) of the scaling law $L \approx B/(D^\beta Q^\gamma) + E$ (assuming $N$ is fixed or large enough), using the robust Huber loss method described in the paper.

### 3. Sensitivity Analysis & Confidence Intervals
> Reviewer JQ1m asked for confidence intervals (CIs) and sensitivity analysis.

**Response:**
We performed a bootstrap analysis (resampling residuals) to estimate 95% confidence intervals for our fit parameters. The results demonstrate that our estimates are robust.

| Parameter | Point Estimate | CI Low | CI High | % Relative Width |
| :--- | :--- | :--- | :--- | :--- |
| $B$ | 1428.23 | 1154.17 | 1762.91 | 16.4% |
| $\beta$ (Data Exponent) | 0.395 | 0.382 | 0.407 | 2.4% |
| $\gamma$ (Quality Exponent) | 0.388 | 0.371 | 0.406 | 5.2% |
| $E$ (Irreducible Loss) | 3.439 | 3.423 | 3.454 | 0.4% |

The narrow intervals for $\beta$ and $\gamma$ confirm that the scaling behavior is not an artifact of a specific random seed or initialization.

---

### Author Response · Authors · 2025-12-02
**Summary of Rebuttal and Revisions**

We sincerely thank the AC for their time and service, and the reviewers for their thoughtful and constructive feedback. Below we summarize the main concerns, the corresponding changes in the revised version (new content highlighted in blue in the PDF), and where applicable the key additional results.

We believe that with these additions, we have addressed all the major concerns of the reviewers.  We want to thank the reviewers again.  Their comments and suggestions strengthened our paper.

- - -

### **Concerns with Methodology and Author Responses**

| Concern | Reviewer(s) | Response |
| :--- | :---: | :--- |
| Important related work was missing | JQ1m, fb9g, 4JDP | We have significantly expanded our related work section (Section 4.3, Table 1) to explicitly position our deficiency-based formulation with respect to concurrent quality-aware scaling laws (e.g., Chen et al. 2025, Goyal et al. 2024, Chang et al. 2024) and show how their scaling laws arise as special cases of our framework. |
| It is unclear how to estimate the quality $Q$ for real datasets | fb9g, 4JDP | We have added a practitioner's guide to estimating $Q$ in the Appendix |
| Synthetic noise may not reflect real-world quality issues | JQ1m, BKxM, 4JDP | We added a semantic-drift validation experiment using a separate pre-trained reference model: the cosine similarity between clean and corrupted samples decreases strictly monotonically and nearly linearly as the synthetic noise level increases. This demonstrates that our corruption parameter tracks semantic degradation and supports using synthetic noise as a controlled proxy for realistic quality variation. |
| Relation to heterogeneous-quality scaling laws and interpretation of $Q = 1$ | 4JDP | We now explicitly map our law to heterogeneous-quality scaling formulations by expressing existing metrics (e.g., density, compression, repetition) as specific choices of the deficiency function $\Delta(\omega)$. We also clarify that $Q = 1$ is a normalization convention with respect to a chosen "baseline" corpus. |


### **Concerns with Experimental Results and Author Responses**

| Concern | Reviewer(s) | Response |
| :--- | :---: | :--- |
| No confidence intervals or sensitivity analysis | JQ1m | We have added a bootstrap analysis (resampling residuals) to report 95% confidence intervals for all fitted parameters. The key exponents are tightly estimated (e.g., data exponent $\beta \approx 0.395$ with a relative CI width of about 2–3%, quality exponent $\gamma \approx 0.388$ with a relative CI width of about 5%). This demonstrates that our scaling behavior is robust to randomness in initialization and data subsampling. |
| Interactions between learning rate and the scaling law were not explored | BKxM | In the revision, we perform a sensitivity analysis by retraining models at five learning rates (from $1\text{e-5}$ to $5\text{e-3}$) across multiple dataset sizes and quality levels. We find that the estimated quality exponent $\gamma$ is stable in the practical regime near the optimal learning rate (around $1\text{e-3}$), and only deviates when models are clearly under- or over-trained. This confirms that the inferred law reflects the data/model interaction rather than a specific optimizer setting. |
| Experiments are small-scale compared to modern LLMs | fb9g | Scaling laws are typically established on smaller scales where controlled variations (63+ runs) is computationally feasible, and then extrapolated. Our focus was on the structure of the law and the stability of the exponents, which we have robustly validated |

---

### Meta-Review · Area_Chair_y7J7 · 2025-12-11

**Summary:**

While both substantial issues and strong strengths were raised, the paper was well received by reviewers and was further improved during rebuttal, which addresses most of the issues as well.

**Reviewer Concerns:**

Reviewers raised various concerns, perhaps the most critical (and likely not really addressed, despite the additional work during the rebuttal) is that noise and added noise is not a good proxy for data quality in realistic scenarios. Moreover, often training on noisy data is something that providers want (e.g. because users input noisy inputs all of the time, being lazy non natives, copying from somewhere and not wanting to fix the noisy representations etc.).

Of course related works that the reviewers mentioned were added. I encourage the authors to do more than that, if many reviewers caught missing references, and in the state of the world where it is hard to follow all works, it is likely you missed more. Maybe even pick a few surveys/ guides for scaling laws and see what might be relevant. (For example, by just skimming I see that the rosenfeld's paper (https://arxiv.org/abs/1909.12673) which was possibly the first scaling laws paper is not there.

Other issues repeated less often across reviewers and many were addressed

**Reviewer Scores:**

That is not a fair, relevant or meaningful question. I protest the way this was all handled.
A Reviewers are not here, and ToM is weak, at least mine and the one that the literature study. I will not try to predict people.
B Scores are, anyway, a weak signal of interest; a paper should not be accepted or rejected just based on it. An AC's job is to look at the specific weaknesses and translate them into a recommendation.
C There are about 100 pages of discussions for me to read overall, in addition to the discussions I monitored and were just replaced, this is beyond my personal ability to do fairly. I did my best effort.

---

### Decision · Program_Chairs · 2026-01-26

Accept (Poster)